# High-Purity Preparation of Enzyme Transformed Trans-Crocetin Reclaimed from Gardenia Fruit Waste

**DOI:** 10.3390/plants11030281

**Published:** 2022-01-21

**Authors:** Charng-Cherng Chyau, Chu-Ying Chiu, Hung-Lin Hsieh, David Wang-Chi Hsieh, Chong-Ru Hsieh, Chi-Huang Chang, Robert Y. Peng

**Affiliations:** 1Research Institute of Biotechnology, Hungkuang University, Taichung 43302, Taiwan; onepiece98012@gmail.com (C.-Y.C.); ok1456@sunrise.hk.edu.tw (C.-H.C.); 2Day Spring Biotech Co., Ltd., Taichung 43302, Taiwan; ken520416@gmail.com (H.-L.H.); mark430288@gmail.com (D.W.-C.H.); lawrence5577@gmail.com (C.-R.H.)

**Keywords:** *Gardenia jasminoides* fruit waste (GFW), crocins, *trans*-crocetin, resin adsorption, centrifugal partition chromatography (CPC)

## Abstract

The recovery of physiologically bioactive ingredients from agricultural wastes as an abundant and low-cost source for the production of high value-added mutraceuticlas has been recognized and supported for the commercial interests and sustainable managements. In the extraction of geniposide for the development of natural food colorants from the dried fruits of *Gardenia jasminoides* Rubiaceae, the gardenia fruit waste (GFW) still remaining 0.86% (*w*/*w*) of crocins has always been discarded without any further treatments Until now, there was no simple and effective protocol for high-purity *trans*-crocein (TC) preparation without the coexistence of non-biologically active *cis*-crocein from GFW. We proposed an effective process to obtain the compound as follows. Crocins were extracted firstly by 50% of ethanol in the highest yield of 8.61 mg/g (*w*/*w*) from GFW. After the HPD-100 column fractionation in the collecting of crocins, the conversion ratio of 75% of crocins to crocetins can be obtained from the commercial available enzyme- Celluclast^®^ 1.5 L. The crocins hydrolyzed products, were then separated through the HPD-100 resin adsorption and finally purified with the centrifugal partition chromatography (CPC) in single-step to obtain TC in a purity of 96.76 ± 0.17%. Conclusively, the effective enzyme transformation and purification co-operated with CPC technologies on crocins resulted in a high purity product of TC may be highly application in the commercial production.

## 1. Introduction

Crocins, a group of carotenoid-derived natural products occurring in gardenia fruit (*Gardenia jasminoides* Ellis) and saffron (*Crocus sativus* L.), have long been used as a traditional herbal medicine and a natural colorant [1,2,3]. Saffron is considered the world’s costliest spice, valued at 1500 to 2200 Euro/kg [4], while the gardenia fruit is only cost US$ 1.2 to 2.0/kg from the contractual farms in the study. Up to present, a number of useful chemical constituents have been isolated from *G. jasminoides* (Figure 1) and characterized [2]. The major ones include geniposide (GPS), geniposidic acid (GPSA), genipin (GNP), gardenoside, crocins; and their derivatives [2]. A diversity of its biological activities has been documented, which include antioxidant properties [5], anti-inflammatory [6,7], antidiabetic [8], antidepression [9], improving insomnia [1], anti-hypertension, anti-hyperglycemia, anti-cancer, anti-hyperlipidemia, and Alzheimer’s disease (AD) [10]. GPS alleviated cholestasis liver diseases by regulating the expression of bile salt export pump (BSEP) through farnesoid X receptor (FXR) and nuclear factor erythroid 2–related factor 2 (Nrf2) signaling pathway [11].

Crocins are water soluble compounds. The largest source of crocins and crocetin is from saffron stigma [9], as contrast, the less expensive *G. jasminoides* is seldom used for such a source [3]. A number of methods have been used for isolation of crocins include the solvent partition separation, the classic column chromatographic (CC) method, macroporous resin adsorption and preparative HPLC method. In which, HPLC has been used for separation of picrocrocin, *cis*- and *trans*-crocins (as shown in Figure 2), and safranal [12]. Alternately, a support matrix free separation technology, i.e., the high-speed countercurrent chromatography (HSCCC) has been used to isolate the phytochemicals to replace the HPLC method with high purities [2].

HSCCC is a liquid-liquid partition technology which has successfully isolated geniposide [13], iridoid glycosides and crocetin derivatives from *G. jasminoides* [14]. Recently, Karkoula et al. [15] have conducted a similar purification method on crocins, picrocrocin, and crocetin from saffron using the centrifugal partition chromatograph (CPC) facility and claimed that the facility could be easily scaled-up as an extremely useful tool for the separation of complex mixtures [15]. After the fractionation using macroporous resin (HPD-100), HSCCC is used to isolate crocin-1, crocin-2, crocin-3 and crocin-4 from gardenia fruit by the solvent system (hexane/ethyl acetate/*n*-butanol/water, 1:2:1:5, *v*/*v*) and obtain the purities of 94.1%, 96.3%, 94.1% and 98.9% respectively [14]. In addition, HSCCC has been applied to isolate the rarely reported iridoid glucosides from *Lamiophlomis rotata* (Benth.) Kudo, including shanzhiside methyl ester, phloyoside, chlorotuberside, and penstemonoside [16].

Crocins have been shown to possess various medicinal properties [10]. Worth noting, crocins even at high concentrations (1000 µM) cannot penetrate the intestinal lumen, nor absorbable into the villi unless it has been transformed into deglycosylated TC, which can be absorbed in a dose dependent manner by passive transcellular diffusion within a short time interval over the intestinal barrier [17]. On the other hand, crocetin is able to penetrate in a quite slow process the blood brain barrier (BBB) to reach the CNS [17]. Justified from the pharmacokinetic and pharmacodynamic background has revealed that TC, bearing strong NMDA receptor affinity and channel opening activity, exhibits immense potential for treating the CNS disorders [17]. In another neuroprotective study, TC has been indicated to enhance amyloid-β_(1–42)_ degradation in the monocytes of Alzheimer’s disease patients [18]. Besides, retino-protective influence of ingested saffron is caused by crocetin which reaches the retinal pigment epithelia (RPE)/retina following intestinal absorption [19], and obviously, pharmacologically and therapeutically, the bioactivity of crocins can be attributed to the deglycosylated crocetin. In consideration of real metabolic conditions, the bioactivity research in the cell models, i.e., neuron cells, ocular cells and hepatocytes etc., the crocetin rather than crocin would be pertinently applied in the studies. Therefore, a simple and effective operation for crocetin preparation is of special interest given it wide range of studies in functional food and medicine industries.

In gardenia fruit processing, different parts of fruits contained quite different compositions (Figure 1). Crocins abundantly occur in the gardenia fruit pulps, while iridoids like geniposide and geniposidic acid are prevailing compounds in pericarps and pulps. Crocins pigments including crocin 1 and crocin 2 also occur abundantly in pericarps and pulps [19] (Figure 1). The extraction of geniposide for developing natural food colorants usually leaves behind a tremendous amount of waste pericarps and pulps, causing a primary environmental pollution problem. Moreover, although the crocins are also extracted together with geniposide during the extraction processing, usually the content of residual crocins is still rather high and worth reclaiming.

In previous studies, alkali-mediated conversion of crocins to crocetin method has been applied in the preparation of crocetin [20], until now the information in literature on the enzymatic hydrolysis of crocins in vitro for the preparation of crocetin is very limited [21]. We hypothesized that the commercial cocktails enzymes Celluclast^®^ 1.5 L containing cellulase and β-glucosidase could be applied to carry out the enzyme–catalyzed transformation. At this stage, a moderately high purity crocetin was obtained. The aim of this study was further to improve the purity of TC, we adopted a serial process by combining macroporous resin adsorption and the newly developed centrifugal partition chromatography (CPC) to reach the goal.

## 2. Results and Discussion

### 2.1. Comparison of the Composition between the Dried Gardenia Fruits and GFW

In order to overcome the source of raw materials, we first analyze whether there are any targets in the raw materials that can be used. The proximate compositions in oven-dried gardenia fruits and GFW, including moisture, crude protein, crude fat, ash and carbohydrate were shown in Table 1. The GPS contents in the dried gardenia fruit were 3.18% (*w*/*w*) and the GFW were 0.54%, i.e., most part of GPS was extracted for commercial use. The content of other compositions were rather comparable. However, there was a large difference between the two samples regarding that of GPS and crocins. Considerable amounts of crocins (14.09%, *w*/*w*) (Table 1) were found in gardenia fruit in comparable with the report from Gao and Zhu [22]. The fact is that levels of carotenoids formed in gardenia fruits have been indicated in different maturity stages of fruits, in which the carotenoid pigment content increases rapidly from 3.96 μg/mg to 119.29 μg/mg dry weight in the four development stages from the 8th week to the 36th week after flowering. Nevertheless, the geniposide concentration in gardenias fruits was unchanged drastically from 17.99 to 23.16 μg/mg during the growth stages [22]. After the extraction of geniposide from gardenia fruit for colorant usage, there still existed in 0.86% of crocins (Table 1) in the GFW.

Gardenia fruit contains a number of bioactive compounds particularly rich in iridoid glycosides, of which geniposide is the most abundant [2]. In addition to conferring pharmacological functions, geniposide also has the function as the raw material for producing colorants [23]. Wu et al. [9] have analyzed 34 batches samples of gardenia fruit and found that the content of geniposide in most of gardenia fruit samples were between 3.6% and 4.1% (*w*/*w*). Similarly, a comparable content of 3.18% of geniposide was determined in this study (Table 1). In commercial production of geniposide from gardenia fruit, the moderate ethanol percentage (40%) was applied to the extraction of geniposide [24], while most of the remaining part was discarded. The main goal of the present work was to use crocins from the GFW for developing a high value-added bioactive compound in replace of the expensive and rare sources of saffron.

### 2.2. The Percent Extractability and Total Content of Crocins in the Dried Gardenia Fruit

Saffron is the main natural source of crocins (crocetin esters), which are water-soluble carotenoid derivatives constituting in a general content ranges between 16–28% in dried stigmas [25]. Because of its rarity and extremely high price of saffron, seeking alternative sources, i.e., the agricultural wastes have extremely high potential value. In this study, ethanol was selected as the solvent for its safety in human consumption and less negative environment impacts [26]. Ethanol at 25–95% was respectively used to extract the crocins from dried gardenia fruit (GF) and gardenia fruit waste (GFW). The optimum ethanol concentration for extraction of crocins from the intact dried gardenia fruit powder was determined to be 50% with yield of 25.63 ± 2.73%, and the total content of crocins was 14.09 ± 1.02 mg/g (Table 2). Thus, the 50% ethanol was applied to the extraction of crocins in the GFW, which has been extracted out geniposide and crocins, the raw material of colorant production, by using the solvent extraction of 40% ethanol. Interestingly, there was 8.61 mg/g of crocins still remained in the GFW (Table 1). The effect of ethanol on the extraction efficiency of crocins was similar to that the extraction using Liquid CO_2_ under the ethanol modifier addition of 50 or 80% ethanol [27]. It was speculated that the moderate alcohol % used in the extraction of geniposide might incapable in the total extraction of crocins.

To find a suitable ethanol concentration in the extraction of crocins on GF and GFW, four ethanol concentrations such as 25, 50, 75 and 95% ethanol were investigated synchronously. The yield and total content of crocins obtained by different ethanol concentrations in extraction of GF and GFW are shown in Table 2. The results presented that the extraction using 50% ethanol showed significantly potential for extracting crocins than the other ethanol concentrations whether in GW or GFW. The ethanol concentration in the water is beneficial to the increase of the extraction yield which is attributed to moderate polarity of solvent on the nature of crocins., However, too much alcohol concentration is not conducive to the extraction of crocins.

### 2.3. The Optimum Ratio of Enzyme to Crocins Required for the Conversion to Crocetin

After the presence of crocins in 50% ethanol extracts being confirmed by HPLC/MS, the next step was to convert the crocins to crocetin by using enzymatic hydrolysis method to increase the yield of TC. Figure 3 reveals that the ratio 2:1 (crocins solution/enzyme, *v*/*v*) was superior to the other two ratios in releasing TC, liberating more than 80% of theoretical value after 16 h reaction at 50 °C, pH 5.0. The data show that Celluclast 1.5 L contained side activities capable of degrading the water-soluble glycosyl esters, including crocetin di (β-gentiobiosyl) ester. Commercial enzyme cocktails from Novozyme Celluclast 1.5 L have been widely used as sources of endo- and exoglucanases, and β-glucosidase, respectively, for enzymatic hydrolysis [28]. In the study, we have demonstrated that the enzymes Celluclast 1.5 L are applicable for the hydrolysis of crocins to the target of TC.

### 2.4. The Optimum Reaction Time Required for the Conversion of Crocins into Crocetin

Celluclast^®^ 1.5 L has been shown a good tolerance to glucose at pH 3.0 and a low Km value [29]. These characteristics were supposed to be capable for application in the deglucosylation of crocins in the study. The reaction system in composed of crocins (10 mg/mL) and Celluclast^®^ 1.5 L at a ratio of 2:1 (*w*/*w*) was incubated at 50 °C for 24 h. The results showed that Cellclast^®^ 1.5 L acted effectively over a 24 h incubation time against the crocins, leading to an approximate of 75% conversion in a rising near to plateau at 16 h (Figure 4). Figure 4 showed the conversion levels with the increasing tendency in a time-dependent manner. In the present study, the two new peaks appeared at 24.92 min (TC) and 25.70 min (*cis*-crocetin) (Figure 5, bottom panel) were significantly presented to indicate the hydrolysis activity of the commercial enzymes. However, a limited hydrolysis on *cis*-2-gg (peak at 23.87 min, <15.1%) was encountered in the study. Taken together, the optimum reaction system was found to be the one with a ratio of crocins/Celluclast^®^ 1.5 L = 2:1 (*w*/*w*) and incubated at 50 °C for 16 h (Figure 4). The observation indicates the Celluclast^®^ 1.5 L enzymes displayed high stability. In a study of the heat stability of enzymes, the Celluclast 1.5 L suffered only an 18% decrease in its concentration at 50 °C after 4 days, demonstrating remarkable stability at high temperature [28].

**Figure 4 plants-11-00281-f004:**
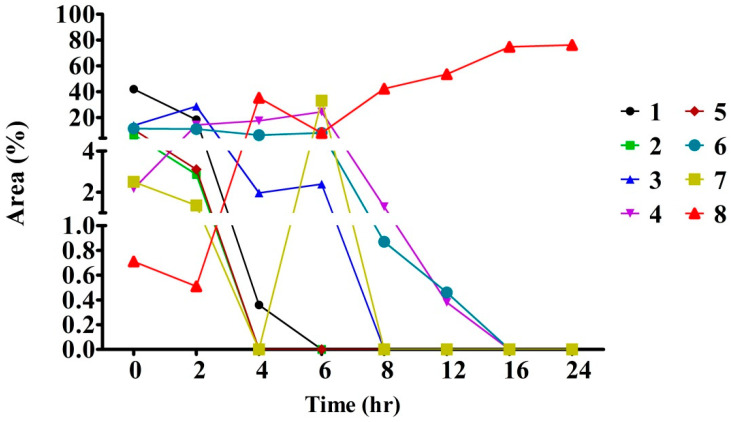
Enzymatic transformation of *trans*-crocetin during a 24 h incubation of crocins extracts from gardenia fruit waste using Celluclast^®^ 1.5 L enzymes. Crocins extracts (2 mM) were reacted with 1.0 mL of Celluclast^®^ 1.5 L enzymes in 10 mL of 100 mM citrate buffer (pH 5.0) at 50 °C. Data presented the averages of 2 experiments as determined by HPLC analysis detected at 440 nm. A description of the numbers is given in Table 3.

### 2.5. Adsorption and Desorption of Hydrolyzed Products via Macroporous Resin HPD-100

Separation and enrichment of TC from the enzyme hydrolyzed products was performed by using HPD-100 resin according to previous report [32]. After that, we subjected the hydrolyzed products to HPLC-ESI/MS analysis. Figure 5 shows the chromatograms of prepared crocins at 0 time and the hydrolyzed products after the bioconversion reaching the plateau of TC production at 16 h, respectively with Celluclast^®^ 1.5 L. As can be seen in the chromatograms, most of the crocins were hydrolyzed to the TC (peak 8) and a small amount of *cis*-crocetin (peak 9). Unexpectedly, there was a compound (peak 7), identified as *cis*-2-gg-crocin, still existed in the hydrolyzed products. However, the purity of 92.16 ± 5.24% of TC was obtained at this stage before the further purification by CPC.

### 2.6. Purity of TC Further Improved by Centrifugal Partition Chromatography (CPC)

CPC processing is based on the solutes separated in different distribution coefficients (*Kd*) between two immiscible solvent systems and the ratio of solvent system is adjusted to get the *Kd* value of target analyte between 0.5 and 2.0 [33]. In this study, the *Kd* values of TC and *cis*-crocetin were performed in four solvent systems as shown in the Table 4. Results showed that the optimum proportion of the two-phase solvent system ethyl acetate/*n*-butanol/water (containing sodium carbonate 0.3 M) = 2.5/2.5/5 (*v*/*v*/*v*, decending mode) was found to be the most suitable for CPC separation of TC (*K_d_* = 1.4) and *cis*-crocetin (*K_d_* = 0.7) (Table 4). The modified systems with water containing 0.3 M Na_2_CO_3_ were applied for the first time in CPC for purifying TC in the study. In fact, previous study has indicated that the addition of the inorganic salt resulted in a very slight decrease in K values and thus a shorter separation time [34]. Thus, as shown in the Figure 6, the applied systems in the study enabled to reduce the separation time of TC to 90 min in a better separation effectiveness than the previous HSCCC processing in consuming about 8 h for collecting TC [35].

### 2.7. The Yield and Purity of TC Post CPC Treatment

The pre-purified crocetins (100 mg/5 mL) from macroporous resin HPD-100 separation were loaded into the CPC system. After separation, the collected tubes from No. 26 to No. 50 (Figure 5) were combined and evaporated in vacuo to obtain the TC at a yield of 95.25 ± 3.16% in a purity 96.76 ± 0.17%. Figure 6 illustrates the CPC–UV-Vis chromatogram with two fractions (retention time 10–12 and 30–60 min, respectively) separated in less than 90 min. The extremely high yield of CPC purification has been indicated that the chromatographic technique of CPC does not need solid support, i.e., resins as a stationary phase, therefore provides many advantages over conventional chromatography, including the none irreversible adsorption onto column materials and low risk of sample denaturation [36].

### 2.8. The Identification of trans-and cis-Crocetin in CPC Purified Products

Finally, under the optimized condition of two-phase solvent system of CPC, *trans*- and *cis*-crocetina were the only remained constituent in the purified sample (Figure 6). It can be seen from the results that the CPC can obviously remove the minute compound *cis*-2-gg-crocin observed from Figure 5 (bottom panel) and Figure 7 (top panel). To resolute the next question in discriminating *trans*- and *cis*-form of crestion, the UV-Vis and MS spectra were compared with each other. As shown in Figure 7 (botom panel), an additional absorption band around 320 nm was observed in the *cis*-crocetin, which has been indicated as an important indicator for distinguishing from the *trans*-form [37]. Unfortunately, the MS method failed to achieve the differentiation between the *cis*- and *trans*-crocetin stuctures from the positive ion mode of MS/MS experiments (Figure 8). The product ions produced from the molecular ion of *m*/*z* 329 provided a similar fragmentation pattern between *cis*- and *trans*-form, let the identification of the two geometric isomers fail to identify correctly. In short, ESI-MS spectra provided information of the molecule weight and UV-Vis spectra presented the absorption of the *trans* conjugated double bonds. The combination method of UV-Vis absorption and MS spectrum offers rapid and accurate identification of *cis*- and *trans*-crocetin.

## 3. Materials and Methods

### 3.1. Materials

The commercial enzyme source-Celluclast^®^ 1.5 L (batch No. 04070012) obtained from Novozymes (Bagsværd, Denmark) was applied for the hydrolysis of crocins in the production of crocetin. The macroporous resin HPD-100 was purchased from Cangzhou Bonchem Co., Ltd. (Cangzhou, Hebei, China). Crocin (crocetin digentiobiose ester) and geniposide were from Sigma Aldrich (St. Louis, MO, USA). 

### 3.2. Source of the Gardenia Fruit Waste (GFW)

The fresh fruits of *G*. *jasminoides* Ellis were harvested in December, 2018 at the contractual farms located at Chiayi (120E27′00”, 23N36′00”), Taiwan. After air- and oven (50 °C)-dried, and pulverized (40 mesh), the main constituent geniposide was extracted firstly with 40% ethanol (1:10, *w*/*v*) in triplicate according to the commercialization process. The residue, after oven-dried at 50 °C, was designated as gardenia fruit waste (GFW) and used as the raw material in this study.

### 3.3. Proximate Analysis of the Dried Gardenia Fruits and GFW

The Association of Official Analytical Chemists [38] methods of analysis was used to determine the moisture, protein, fat, ash and carbohydrate contents of the oven-dried gardenia fruits and GFW at 50 °C for 8 h.

### 3.4. Determination of Geniposide and Crocins in the Dried G. Jasminoides Fruit Powders

To 1.0 g of untreated dried *Gardenia jasminoides* fruit powders 10 mL of ethanol at different concentration ranging from 50–90% was added, agitated at 120 rpm for 15 min, centrifuged at 3000× *g* for 5 min, and filtered through a 0.45 μm PVDF filter paper. The extraction was repeated for three times. The filtrates were combined and concentrated in vacuo to get the crude geniposide extract. To the residue 10 mL of combined solvent water: ethyl acetate (1:1, *v*/*v*) was added, agitated at 120 rpm for 15 min, and left to stand until phase separated. The lower layer was collected, evaporated and subjected to HPLC analysis. The crocin standard (crocetin digentiobiose ester, Sigma) was used to establish the calibration curve, from which the purity was calculated.

### 3.5. Reclaim of Crocins from the Gardenia Fruit Waste (GFW)

Gardenia fruit waste (GFW) (10 g) was suspended in and extracted with 40 mL of ethanol at different concentration (50–90%), ultrasonicated for 15 min at different temperature, and filtered through 0.45 μm PVDF filter paper. Triplicate experiments were performed. The three filtrates were combined and subjected to rotary evaporator heated in a water bath at 40 °C and under reduced pressure until a viscous paste obtained. The viscous condensate was re-dissolved in 40 mL of deionized water and was transferred into a separation funnel. Ethyl acetate (40 mL) was added, agitated vigorously and left to stand at ambient temperature for 1 h to facilitate the phase separation. The ethyl acetate layer was separated. The aqueous layer was re-concentrated in the rotary evaporator heated on a water bath at 40 °C and under reduced pressure (50 mmHg) to remove the remaining ethyl acetate. The concentrate containing crude crocins was re-dissolved in methanol (1 mg/mL, *w*/*v*) and filtered through a 0.22-µm PTFE membrane and subjected to HPLC for analyzing the chemical compositions.

### 3.6. HPLC Analysis and LC/MS Identification on the Compositions of Crude Crocins Extract

The quantitative analysis was carried out in an Agilent 1200 HPLC system connecting a precolumn [SecurityGuard C18(ODS) 4 mm × 3.0 mm i.d., Phenomenex Inc., Torrance, CA, USA] and an analytical C18 column (ZORBAX Eclipse Plus C18, 2.1 × 100 mm, 1.8 μm, Agilent, CA, USA) in a column oven set at 35 °C. The elution mobile systems used were (A) 0.1% formic acid in deionized water and (B) 0.1% formic acid in acetonitrile. The entire course of programmed gradient elution was carried out as follows: 0–1 min, isocratic with 5% B; 1–4 min, with 5–10% B; 4–20 min, with 10–30% B; 20–25 min, with 30–95% B; 25–30 min, isocratic at 95% B and returning to initial conditions in 10 min. The HPLC was operated at a flow rate of 0.3 mL/min, injection volume 10 μL. A diode array detector (DAD) was monitored at 254, 280, and 440 nm when scanning from 210 to 600 nm.

The HPLC-triple quadrupole mass spectrometry system (The Agilent 6420, Santa Clara, CA, USA) equipped with electrospray ionization (ESI) interface was used to confirm the compositions of extracts for which compounds having been eluted and separated. The nitrogen gas acted with both functions, one as the drying gas controlled at a flow rate of 9 L/min, the other acted as a nebulising gas operated at 35 psi. The drying gas temperature was 325 °C, and a potential 3500 V was applied across the capillary. The fragmentor voltage was set at 125 V, and the collision voltage, 25 V. The quadrupole 1 filtered the calculated m/z of each compound of interest, while quadrupole 2 scanned for ions produced by nitrogen collision of these ionized compounds in the range100–1000 *m*/*z* at a scan time of 200 ms per cycle. Mass data were acquired in negative and positive ionization mode. The identification of separated compounds was carried out by comparing their mass spectra provided by ESI-MS and ESI-MS/MS as described in literatures [30,31,39].

### 3.7. Optimum Reaction Time for Conversion of Crocins to Crocetin by Celluclast^®^ 1.5 L

Considering the screening of commercial enzyme preparations containing β-glucosidase activity, the Celluclast^®^ 1.5 L was chosen to apply the hydrolysis study of crcins from nine commercial products as described previously [29]. In brief, the crocins were hydrolyzed using Celluclast^®^ 1.5 L at 2:1 ratio (*w*/*w*) as per the instructions by the manufacture. In which, crocins and Celluclast^®^ 1.5 L were dissolved in 0.1 M citrate buffer (pH 5.0) and incubated at 50 °C for different reaction time at 0, 2, 4, 6, 8, 12, 16 and 24 h, respectively. The enzymatic reaction was performed at dark room under nitrogen atmosphere and gentle stirring (150 rpm).

### 3.8. Optimum Ratio of Substrate to Enzyme for Conversion of Crocins to Crocetin by Celluclast^®^ 1.5 L

The concentrations used in this study were chosen on the basis of preliminary experiments. Crocins were respectively treated with Celluclast^®^ 1.5 L at a ratio 10:1, 5:1, and 2:1 (*w*/*w*) to test for the optimum condition to convert crocins into crocetin. In brief, the reaction mixture was incubated at 50 °C for 16 h. After the reaction was completed, an equal volume of ethyl acetate was added, agitated for 25 min, and left to stand at ambient temperature until phase separation. The ethyl acetate layer was collected and evaporated to dryness. The residue was re-dissolved in 2 mL ethyl acetate and subjected to HPLC analysis.

### 3.9. Macroporous Resin Adsorption

To 10 mL of crude crocetin solution obtained from the hydrolysis of crocins at 16 h was filtered through 0.45 μm filter paper, and was subjected to the macroporous HPD-100 resin adsorptive column. The adsorbed crocetin was eluted with 95% ethanol. The fractions that contained crocetins were combined and concentrated under reduced pressure to obtain pre-purified crocetin. By HPLC analysis, the yield was calculated.

### 3.10. Purification with Centrifugal Partition Chromatography

After treated with the macropore resin HPD-100 adsorption chromatography, the pre-purified crocetin products were further purified by using centrifugal partition chromatographay (CPC, Spot Prep II, Armen Instrument, Saint-Avé, France), under the two-phase solvent system. In brief, the ethyl acetate-*n*-butanol-water (containing sodium carbonate 0.3 M) solvent systems in different proportions were prepared. Ten mg of crude sample was added to a 10-mL tube, and then, 5 mL of each phase of the pre-equilibrated two-phase solvent system was added and vigorously shaken. After the two-phase samples were thoroughly equilibrated, 20 μL of each phase from the upper- and the lower phases was collected and subjected to HPLC analysis. The peak area representing TC present in each phase was measured respectively, and the ratio obtained was designated as the partition coefficient, *K_d_*. The well recommended technical parameter for *K_d_* is 0.5 to 2.0 [40]. From the above different solvent systems the one having optimum proportion was selected. The upper phase solution was selected as the stationary phase, with the lower phase solution as the mobile phase. The upper phase solution was pumped at a flow rate of 30 mL/min into the CPC, left until stabilized, and the centrifugation was set at 1600 rpm. The mobile phase was fed at a flow rate 8 mL/min and the outlet stream was checked for correct phase separation. On viewing apparent phase separation of the outlet stream, the crude sample (100 mg/5 mL) was injected. The fraction of outlet stream exhibiting maximum absorption at 440 nm (a typical UV-Vis absorption wavelength of crocetins) was collected, 5 mL in each tube. The fractions having absorption maxima at 440 nm were examined with HPLC, and all those fractions containing TC were pooled, concentrated and checked for its component and purity with LC/MS.

### 3.11. Statistical Analysis

Data of mean ± SD manner were presented in the study and further analyzed by soft GraphPad Prism Program (GraphPad, San Diego, CA, USA). One way analysis of variance (ANOVA) was used for analysis of variations in each group. Tukey’s post hoc test was used for analysis of the significance of difference among the means. A confidence level *p* < 0.05 was considered to be statistically significant.

## 4. Conclusions

In this work, we prepared TC from crocins in an enzymatic hydrolysis method and a simple and effective purification procedure from economic available GFW by using one-step CPC purification to obtain a high purity TC (96%) in a yield of 5.03 mg/g. Compared with the conventional purification methods, such as HPLC and column chromatography, the proposed CPC method is scalable under a variety of operating modes through a few simple experiments to reach the purification goals. Another area of future work is searching for the suitable solvent system for the separation and purification of bioactive TC from the co-existed ***cis***-crocetin, owing to the similar molecular sizes and close solubilities. In view of the novel drug of *trans* sodium crocetinate (TSC) has been demonstrated in clinical benefit in the treatment of COVID-19-related hypoxemia [41], the purification method and the therapeutic use of *trans*-crocetin would be urgent and great potential.

## Figures and Tables

**Figure 1 plants-11-00281-f001:**
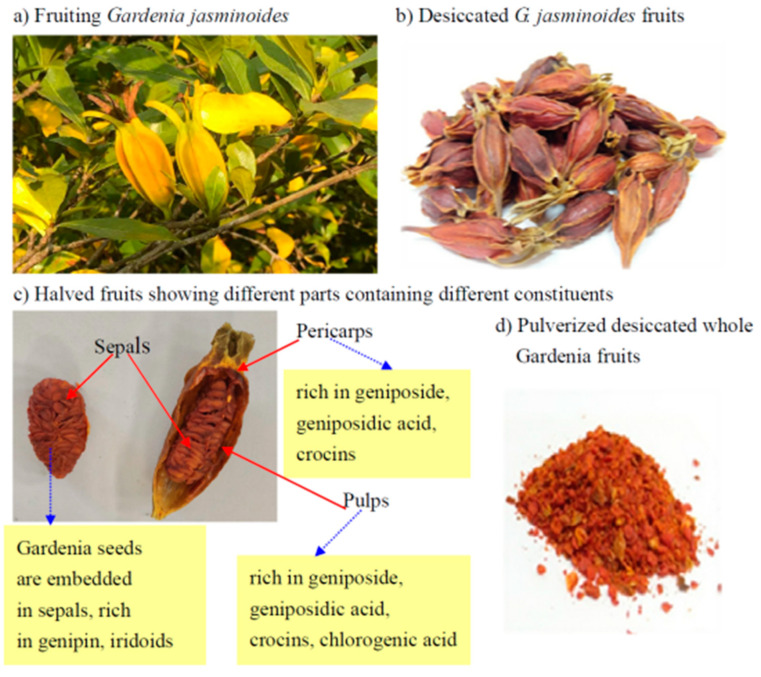
*Gardenia jasminoides* fruits and constituents. (**a**) fruiting *G. jasminoides,* (**b**) dried fruits, (**c**) different parts contain different constituents, (**d**) pulverized whole fruits for extraction and isolation of geniposide and crocins.

**Figure 2 plants-11-00281-f002:**
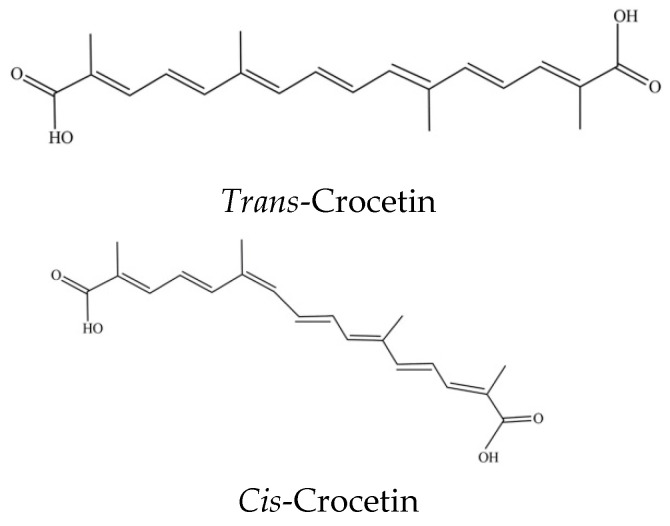
The chemical structures of *trans*- and *cis*-crocetin.

**Figure 3 plants-11-00281-f003:**
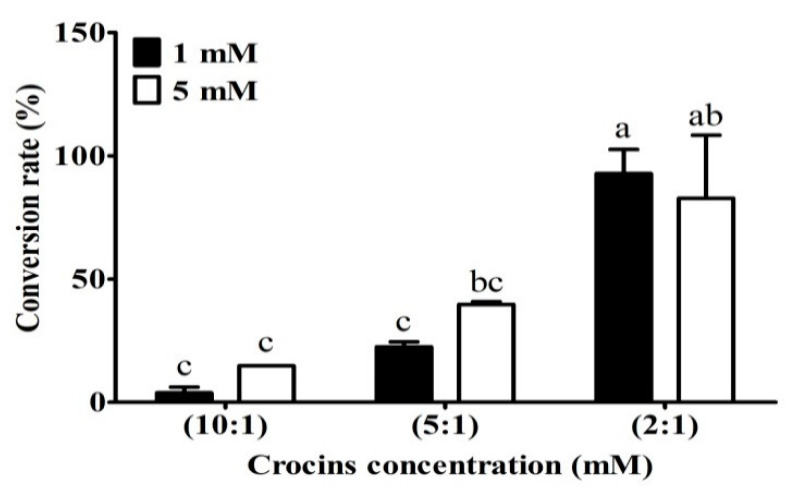
Conversion rate of different concentrations of crocins in enzymes. Crocins (at 1 mM and 5 mM, dissolved in 100 mM of citrate buffer, pH 5.0) and Celluclast^®^ 1.5 L at ratios 10:1, 5:1; and 2:1 (*v*/*v*) were incubated at 50 °C. for 16 h. Different letters in the group indicate significant differences according to the Tukey’s multiple comparison test (*p* ≤ 0.05).

**Figure 5 plants-11-00281-f005:**
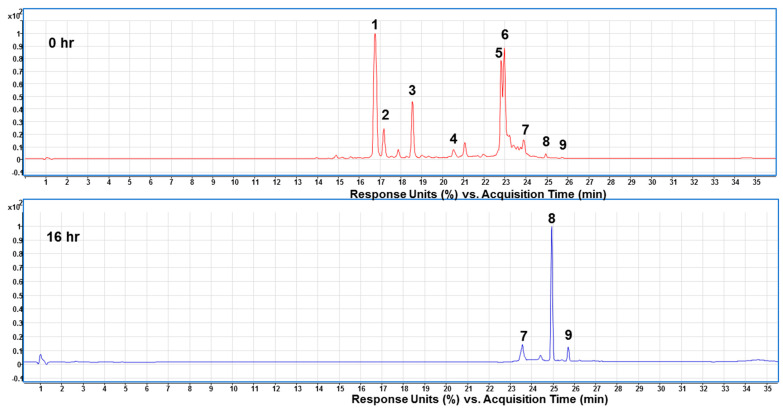
HPLC profiles for the bioconversion of crocins extracts using Celluclast^®^ 1.5 L enzymes detected at 440 nm in the incubation time of 0 and 16 h. HPLC parameters: C18 column (1.8 µm, 2.1 × 100 mm; Agilent Eclipse Plus) column oven 35 °C, mobile phases (**A**) 0.1% formic acid in water and (**B**) acetonitrile (containing 0.1% Formic acid) at 0.3 mL/min. DAD: 440 nm. Peak numbers are referred to Table 3.

**Figure 6 plants-11-00281-f006:**
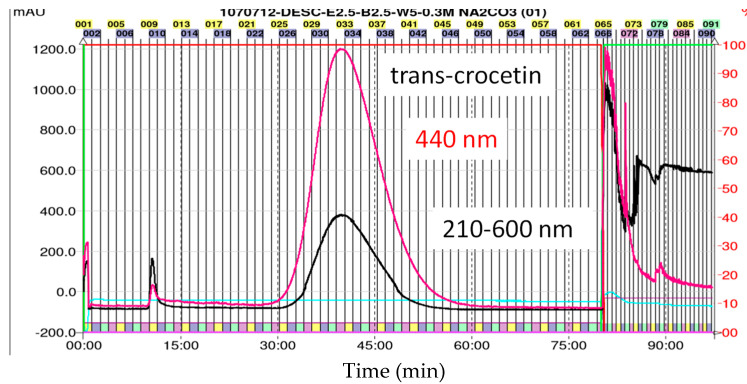
Centrifugal partition chromatography (CPC) obtained from the two-phase solvent system ethyl acetate/*n*-Butanol/0.3 M Na_2_CO_3_ in water (2.5:2.5:5). Red line indicated the absortion wavelength at 440 nm for the detection of *trans*-crocetin. Tubes from No. 26 to No. 50 (time 31.25–60 min) were combined and evaporized in vacuo to obtain the target *trans*-crocetin.

**Figure 7 plants-11-00281-f007:**
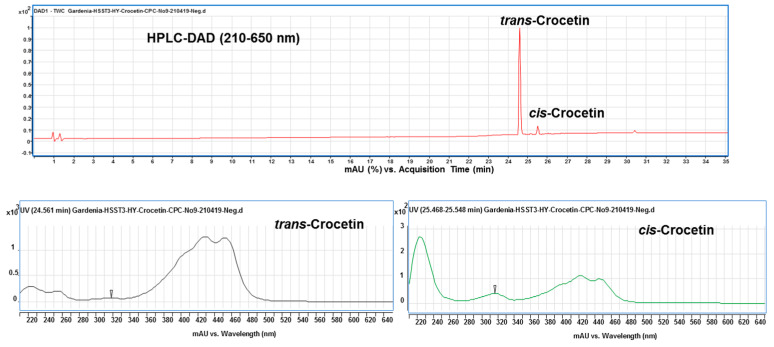
HPLC chromatogram (top panel) and UV-Vis spectrum (bottom panel) of *trans*- and *cis*-crocetin obtained from the CPC purification. *Cis*-crocetin showed a significant absorption peak at 316 nm different from that of *trans*-crocetin without the wavelength absorption.

**Figure 8 plants-11-00281-f008:**
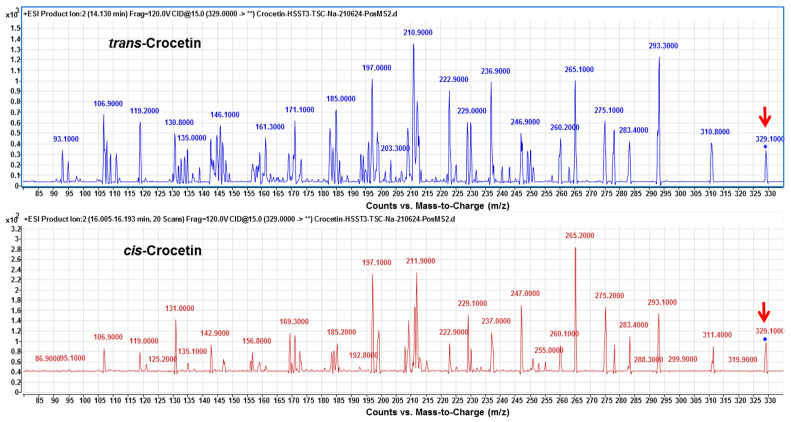
The MS spectra of *trans*- and *cis*-crocetin from HPLC-ESI (+)-MS/MS analysis. Fragmentations from collision- induced dissociation of molecular ion at *m*/*z* 329 (as indicated by the red arrow) were produced in the positive ion mode of electrospray ionization. The MS fragmentations of *trans*- and *cis*-crocetin did not illustrate a significant difference between each other.

**Table 1 plants-11-00281-t001:** Proximate composition, geniposide, and crocins analyses in the dried gardenia fruits and gardenia fruit wastes.

Term	Content % (*w*/*w*) ^a^	
	Dried Gardenia Fruit	Gardenia Fruit Waste
Moisture	10.12 ± 0.02	5.85 ± 0.02
Crude protein	6.36 ± 0.22	8.89 ± 0.05
Crude fat	19.15 ± 0.26	18.89 ± 1.25
Ash	4.82 ± 0.13	5.24 ± 0.26
Carbohydrate	59.55 ± 0.19	66.92 ± 1.20
Geniposide	3.18 ± 0.47	0.54 ± 0.08
Crocins	14.09 ± 1.02	0.86 ± 0.01

^a^ Each value was expressed as mean ± SD of triplicate samples.

**Table 2 plants-11-00281-t002:** The extractability and total content of crocins from the dried gardenia fruit (GF) and gardenia fruit waste (GFW).

Ethanol (%)	Yield of Extract (*w*/*w* %) ^1^ GF GFW		Total Content of Crocins (mg/g DW) ^1^	
			GF	GFW
25	19.90 ± 1.83 ^b^	10.28 ± 0.99 ^b^	10.28 ± 0.99 ^b^	4.15 ± 0.52 ^b^
50	25.63 ± 2.73 ^a^	14.09 ± 1.02 ^a^	14.09 ± 1.02 ^a^	8.61 ± 0.63 ^a^
75	14.26 ± 0.41 ^c^	9.20 ± 0.34 ^b^	9.20 ± 0.34 ^b^	6.13 ± 0.41 ^a^
95	10.77 ± 0.98 ^c^	4.09 ± 0.30 ^c^	4.09 ± 0.30 ^c^	1.33 ± 0.20 ^c^

^1^ Values with the different letters in the column are significantly different (*p* < 0.05) by Tukey’s multiple comparison test.

**Table 3 plants-11-00281-t003:** Identification of crocins relative compounds in raw materials and enzyme hydrolyzed products of GFW by using HPLC-DAD-ESI-(+)-MS spectrometry.

Peak No. ^1^	Retention Time (min)	λmax (nm)	Molecular Weight	Molecular Ion (*m*/*z*)	Fragmentation (*m*/*z*) ^2^	Identified Crocins
1	16.65	438, 466	976.96	999 [M + Na] ^+^	329, 311, 999	*trans*-4-GG ^3^
2	17.13	440, 464	976.96	999 [M + Na] ^+^	635, 473, 999	*cis*-4-GG ^3^
3	18.51	444, 464	814.82	837 [M + Na] ^+^	327, 837, 311	*trans*-3-Gg ^3^
4	20.50	438, 460	652.26	675 [M + Na] ^+^	675, 323, 346	*trans*-2-G ^3^
5	22.74	436, 460	976.96	999 [M + Na] ^+^	721, 311, 999	*cis*-4-ng ^3^
6	22.91	442, 460	652.26	675 [M + Na] ^+^	675, 311, 329	*cis*-2-G ^3^
7	23.87	430, 452	652.26	675 [M + Na] ^+^	675, 228, 329	*cis*-2-gg ^3^
8	24.92	426, 450	328.40	329 [M + H] ^+^	329, 311, 293	*trans*-Crocetin ^4^
9	25.70	424, 444	328.40	329 [M + H] ^+^	311, 329, 293	*cis*-Crocetin ^4^

^1^ Peak numbers are referred to Figure 3 and Figure 4. ^2^ The major fragment ions are ranked in the order of intensity. ^3^ The tentatively identification of crocins has been reported by Suchareau et al. [30] and Bharate et al. [31]. Namely, G refers to gentiobiose and g, to glucose. ^4^ The enzyme hydrolyzed products.

**Table 4 plants-11-00281-t004:** The partition coefficient of *trans*-and *cis*-crocetin between the two-phase solvent system.

Two-Phase Solvent System	Ratio (*v*/*v*/*v*)	*K_d_*_1_^a^ (*Trans*-Crocetin)	*K_d_*_2_^a^ (*Cis*-Crocetin)
ethyl acetate-*n*-Butanol-water *	1:4:5	4.4	13.8
ethyl acetate-*n*-Butanol-water *	2:3:5	2.7	2.2
ethyl acetate-*n*-Butanol-water *	2.5:2.5:5	1.4	0.7
ethyl acetate-*n*-Butanol-water *	3:2:5	0.8	0.6
ethyl acetate-*n*-Butanol-water *	4:1:5	0.1	0.1

* The two-phase solvent system was composed of ethyl acetate/*n*-butanol/H_2_O (containing 0.3 M Na_2_CO_3_). ^a^
*K_d_*: distribution coefficients in descending mode.

## Data Availability

All data generated or analysed during this study are included in this published article.

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
