# Peer review of "High-Purity Preparation of Enzyme Transformed Trans-Crocetin Reclaimed from Gardenia Fruit Waste"

_plants, 2022, doi:10.3390/plants11030281_

Round 1

Reviewer 1 Report

The authors submit a manuscript titled "High-Purity Preparation of Enzyme Transformed Trans-Crocetin Reclaimed from Gardenia Fruit Waste". The work presents an interesting topic and the results are of interest. The methods of purification and the use of enzymes are topics that are always much debated in the scientific community. However, the work should be deeply revised both for the form English often not correct and difficult to understand than for the presentation of the results. Here are some suggestions / corrections to improve the quality of work.

The abstract should be completely rewritten highlighting the main results of the work and highlighting the purpose of the work with its novelties, it is not necessary to report the analytical procedure.

LN 49-55 :” A number of methods have been used for iso- 50 lation of crocins include the common solvent extraction, the solvent partition separation, the classic column chromatographic (CC) methods like reversed-phase-, polyamide-, or Sephadex-, macroporous resin-, prep-HPLC eluting with methanol, acetonitrile or water in different ratio, and certain new technologies like high-speed countercurrent chromatography (HSCCC) “, this sentence seems too long and poorly written, please improve the English form

Ln 72-74: “The combination use of macroporous resin (HPD-100) column separation and HSCCC, different purities of 91.7% to 98.9% of crocin 1and crocin 4 were obtained under the processing “please rewrite this sentence more clearly”

525-527: “TC has been indicated with the potential therapeutic effects on the neuro- degenerative diseases. However, TC is currently prepared from the rare and expensive saffron and is hardly to get a high purity of product.”, please, delete this sentence, t is not suitable for a conclusion section.

Figures;

Fig. 5: please, re-organize the caption of this figure clearly indicating on the graph the various lines with their meanings

Fig 7 should probably be cured by showing the molecular peak and the various fragments. Only in this way can a discussion be organized on the relative differentiation between the two spectra. Eventually a table with the composition of the various fragments with the relative characteristics, could be useful. In the way it is reported it does not provide any useful information

Author Response

Enclosed please find our revised manuscript entitled "High-Purity Preparation of Enzyme Transformed Trans-Crocetin Reclaimed from Gardenia Fruit Waste", which has been revised in according to the reviewer's comments.

All of the revised text has been highlighted in red with the "Track Changes” function according to your guidance.  

We have answered the questions from the two reviewers’ comments item-by-item as the attached file.

Reviewer 2 Report

Authors:

The manuscript on High-purity preparation of enzyme transformed trans-crocetin from the selected plant waste represents an interesting approach to an alternative source of the plant products so far isolated from another but expensive plant source. The selected method, centrifugal partition chromatography for obtaining the target plant product represents a relatively cheap and reliable technique.

Nevertheless, there are several points in this manuscript that need a more detailed investigation and explanation.

(a) It is a certain disadvantage of the manuscript that the structure of the target trans-crocetin is not shown, because it would clearly demonstrate that that both trans- and cis-crocetins appear in the nature as conjugates with monosaccharide or oligosaccharide molecules (more likely) or a soluble salts. The fact that the product described as cis-2-gg-crocin (line 257) exists in the mixture of products after the enzyme-mediated transformation is not too surprising: Enzyme-mediated transformations made uder the laboratory conditions proceed very often with not the quantitative yield, or, more likely, the selected enzyme is unable to hydrolyze the crocetin conjugate(s) with oligosaccharides.

(b) Table 3: However, much more important is a question following from the Table 3 that summarizes crocetins in raw material and enzyme hydrolyzed products. Therefore, it is not clear, what are the natural plant products and what are enzyme-mediated hydrolyzed products. I strongly recommend to separate the products into two different table or to make a clear division of the product groups to know which products are naturally found and which are obtained by enzyme-mediated transformation.

(c) Figure 4 gives the HPLC profile of the natural and enzyme-mediated hydrolysates in the time 0 h and 16 h. However, both chromatograms should contain inner reference compound in the same concentration. Based on the present form of chromatograms in Figure 4, it is not clear, if all compounds (1-6) transformed quantitatively into 8 and 9, and if the concentration of the compound 7 stays constant because it is not transformed by the given enzyme [see also a comment in (a)]?

(d) Table 5: Two-phase solvent system is mentioned, however, ethyl acetate / n-butanol / water represents a more complicated system of three solvents. n-Butanol formes a single-phase mixtures both, either with ethyl acetate or with water, and change in the ratios of the single solvents in this 3 solvent mixtures may influence the solubility of the target crocins in them. However, it seems that this solvent system is more favorable for cis-crocetin than for the target trans-crocetin. I recommend the authors to comment on this factor in more details. The quantity of 0.3 M sodium carbonate is not given - please, add these data. Did you study the solubility of trans- and cis-crocetin in the single solvents?

Formal points:

Configuration descriptors cis and trans should always be written in italics.

Lines 187-194: The line numbers interfere with the text in Table 2.

Line 238: Correct English grammatics, please.

Why number of words or their parts are written in the red color?

In summary, I recommend to modify the manuscript accordingly before it may be accepted for publication after a major revision.

Author Response

Enclosed please find our revised manuscript entitled "High-Purity Preparation of Enzyme Transformed Trans-Crocetin Reclaimed from Gardenia Fruit Waste", which has been revised in according to the reviewer's comments.

All of the revised text has been highlighted in red with the "Track Changes” function according to your guidance.  

We have answered the questions from the reviewer's comments item-by-item as the attached file.

Round 2

Reviewer 1 Report

The revision has greatly improved the manuscript which now shows sufficient quality to be published

Reviewer 2 Report

Authors:

The manuscript has been substantially improved in its revised version. I agree with all your answers and connected modifications of the manuscript.

From the formal point of view, I have found that the line numbers 196-203 interfere with the text of the Table 2. It seems to be a technical problem.

I recommend the revised manuscript for publication now.